# Postoperative Course of Reconstructive Procedures in FGM Type III-Proposal for a Modified Classification of Type III Female Genital Mutilation

**DOI:** 10.3390/ijerph20054439

**Published:** 2023-03-02

**Authors:** Uwe von Fritschen, Cornelia Strunz, Roland Scherer, Marisa von Fritschen, Alba Fricke

**Affiliations:** 1Department of Plastic and Aesthetic Surgery, Hand Surgery, HELIOS Hospital Emil von Behring, Walterhöferstr. 11, 14165 Berlin, Germany; 2Desert Flower Center, Center of Colorectal and Pelvic Floor Surgery, Hospital Waldfriede, Argentinische Allee 40, 14163 Berlin, Germany; 3Department of Plastic and Hand Surgery, University of Freiburg Medical Centre, Medical Faculty of the University of Freiburg, Hugstetter Straße 55, 79106 Freiburg, Germany

**Keywords:** FGM, female genital mutilation, clitoris, reconstruction, classification

## Abstract

Background: Reconstruction after female genital mutilation (FGM) has developed from being merely a therapy for complications to addressing body perception and sexuality. However, evidence regarding a direct correlation between FGM and sexual dysfunction is scarce. The present WHO classification provides an imprecise grading system, which makes it difficult to compare present studies with treatment outcomes. The aim of this study was to develop a new grading system based on a retrospective study of Type III FGM, evaluating operative time and postoperative results. Methods: The extent of clitoral involvement, operative time of prepuce reconstruction and lack of prepuce reconstruction, and postoperative complications of 85 patients with FGM-Type III were retrospectively analyzed at the Desert Flower Center (Waldfriede Hospital, Berlin). Results: Even though universally graded by the WHO, large differences in the degree of damage were found after deinfibulation. In only 42% of patients, a partly resected clitoral glans was found after deinfibulation. There was no significant difference in operative time when comparing patients who required prepuce reconstruction and patients who did not (*p* = 0.1693). However, we found significantly longer operative time in patients who presented with a completely or partly resected clitoral glans when compared to patients with an intact clitoral glans underneath the infibulating scar (*p* < 0.0001). Two of the 34 patients (5.9%) who had a partly resected clitoris required revision surgery, while none of the patients in whom an intact clitoris was discovered under the infibulation required revision. However, these differences in the complication rates between patients with and without a partly resected clitoris were not statistically significant (*p* = 0.1571). Conclusions: A significantly longer operative time was found in patients who presented with a completely or partly resected clitoral glans when compared with patients with an intact clitoral glans underneath the infibulating scar. Furthermore, we found a higher, though not significantly significant, complication rate in patients with a mutilated clitoral glans. In contrast to Type I and II mutilations, the presence of an intact or mutilated clitoral glans underneath the infibulation scar is not addressed in the present WHO classification. We have developed a more precise classification, which may serve as a useful tool when conducting and comparing research studies.

## 1. Introduction

Female genital mutilation (FGM) is not only a medical issue, but is also an issue of human rights. FGM is still widely practiced in Sub-Saharan Africa and beyond due to religious and sociological reasons. It has become more conspicuous in recent years because of increasing migration patterns [1,2]. Only a few years ago, the consequences of this practice were entirely unknown to most health professionals since they were not included in medical curricula. As a result, there was great uncertainty as to how to appropriately care for these women. In addition to the absence of knowledge, there was a lack of infrastructure, which made it extremely difficult for FGM patients to find specialized centers or therapists with a deep understanding of the implications of this sociocultural burden [3]. FGM is associated with complex interrelations of cultural and social background, gender identity, body perception, and psychosexual issues. Many of the women, especially after experiences of war, forced marriage, and rape [4,5], are poly-traumatized. Various studies point out that the change in their peer group after migrating to Western countries greatly changes the women’s view on FGM [6,7]. The impact of all these factors on female sexuality, as well as their interactions with the trauma of FGM, are difficult to assess. Thus, apart from surgical interventions, a thorough consultation along with psychosexual counselling are of utmost importance [8,9]. Nevertheless, one must take into account that it is not clear whether the success of the psychotherapeutic counselling depends on the women’s psychological co-morbidities, the extent of the mutilation, or other factors—just as it is unknown in which way these factors influence the success of surgical approaches [10,11]. Various publications have demonstrated a benefit of surgical clitoral reconstruction [12,13,14,15,16]. However, systematic reviews of studies evaluating safety and clinical outcomes illustrate that these studies provide a very low level of evidence, limited by incomplete follow-up, small sample size, non-validated scales, and various confounding factors [4,17,18,19,20,21,22,23]. In particular, these studies do not allow for the pre-selection of appropriate patients for a specific therapy.

Within the literature, little information is available regarding surgical solutions other than simple deinfibulation [13], which only applies to infibulated women, who represent a minority of all women with FGM. In addition, performing only deinfibulation did not seem to meet the demands of the affected patients, who were also looking for an improved clitoral sensation and the restoration of as normal an appearance as possible [4,14].

Thus, the indication for surgery was expanded in order to improve sensitivity, sexual function, and psychological issues such as body perception. Other than general local flap techniques addressing the aesthetic aspect of vulvar reconstruction, the re-elevation of the clitoral stump marked a major milestone in the history of FGM reconstruction [14,23,24].

An improvement in sexual satisfaction after clitoral reconstruction was reported by Thabet and Thabet in 2003, who assessed sexuality of FGM patients before and after clitorolabioplasty using a self-designed questionnaire, showing that sexual desire and arousal was markedly affected in patients with a mutilated clitoral glans and demonstrating the positive effect of clitorolabioplasty in the restoration of sexuality [12]. Foldés et al. performed a re-elevation of the remaining part of the clitoris after releasing it from the retaining ligaments and overlying scar tissue, exposing the clitoris and leaving it for secondary epithelialization. The latter was performed in order to obtain a thin, soft coverage of the clitoris instead of covering it with local flaps, which would potentially reduce its sensitivity. A total of 866 patients (29% of the patients in whom surgery was performed) attended the 1-year follow-up; 97.7% of them reported an improvement, or at least no worsening in pain and clitoral pleasure [14], thus also showing an improvement in sexual satisfaction.

This approach was adopted and modified by various authors in order to improve the functional outcome [23,25]. Moreover, the desire of women with FGM for as normal an appearance as possible was addressed, extending the reconstructive procedure to improve the visual appearance of the clitoris and vulva. These surgical approaches were thus also suitable for women without infibulation, with the surgical possibilities expanding beyond scar release or fistula and pain management [23].

However, extending the indication for surgery beyond deinfibulation and offering procedures such as the re-exposition of the clitoris or the formation of a neo-prepuce has increasingly raised concerns and controversy [4,16,25]. Nevertheless, various genital reconstructive techniques have proven their value in defect reconstruction after tumor surgery or after infectious processes [26]. Whether they offer a sensible solution for the reconstruction of the clitoris or vulva in FGM has not been sufficiently evaluated to date. Furthermore, the necessary evaluation methods, e.g., questionnaires, are not of satisfactory quality, have not been tested regarding their clinical relevance, and have not been translated into relevant languages. Nevertheless, it is assumed, that—apart from the factors stated above—the extent of the mutilation has a significant effect on the residual potential of sexual function. In this case, the extent of the clitoral mutilation is of particular importance. The presence of a scar or neuroma around this sensitive organ may influence sensation, as well as the scope of the surgical approach, which will have at least a medium-term influence on the postoperative course [14,17,27,28]. In this context, it has been shown that even though the majority of women who had undergone surgery reported a favorable outcome, some patients complained about a new onset or worsening of chronic pain, orgasmic dysfunction, difficulties in sexual arousal after surgery, and post-traumatic stress disorders including flash-backs due to the procedure [14,17].

In order to gain robust evidence on potential benefits and harms of reconstructive surgery after FGM, a universally accepted classification is needed, enabling comparison of research studies and treatment outcomes [23]. The present WHO classification divides FGM into four different types, with Type I referring to the removal of the clitoral glans and/or prepuce, Type II referring to the resection of the labia minora and/or clitoral glans with or without resection of the labia majora, Type III referring to infibulations, and Type IV referring to all other harmful procedures such as pricking, piercing, incising, scraping, and cauterization. Type III is further subdivided by the WHO into Type IIIa (removal and repositioning of the labia minora) and Type IIIb (removal and repositioning of the labia majora). However, within this subdivision it is not mentioned as to whether the clitoral glans has been harmed or left intact [29].

Although this classification offers a basis on which to expand, the extent of Type III mutilations is quite complex (Figure 1), which is why we think there is an urgent need for a further subdivision of Type III mutilations, including information about the integrity of the clitoral glans. Based on an evaluation of intraoperative findings of Type III mutilations and its implication for postoperative complications, we have revised the current classification, providing a useful tool to compare postoperative results from future studies.

### 1.1. Current WHO Classification of Female Genital Mutilation

Type I: Also referred to as Sunna or CircumcisionPartial or total removal of the clitoral glans, and/or the prepuce/clitoral hood
oType Ia. Removal of the prepuce/clitoral hood only.oType Ib. Removal of the clitoral glans with the prepuce/clitoral hood.


Type II: Partial or total removal of the clitoral glans and the labia minora, with or without removal of the labia majora.
oType IIa. Removal of the labia minora only.oType IIb. Partial or total removal of the clitoral glans and the labia minora (prepuce/clitoral hood may be affected).oType IIc. Partial or total removal of the clitoral glans, the labia minora and the labia majora (prepuce/clitoral hood may be affected).


Type III: Also referred to as pharaonic circumcision or InfibulationNarrowing of the vaginal opening with the creation of a covering seal. The seal is formed by cutting and repositioning the labia minora or labia majora. The covering of the vaginal opening is done with or without removal of the clitoral prepuce/clitoral hood and glans (Type I FGM).
oType IIIa. Removal and repositioning of the labia minora.oType IIIb. Removal and repositioning of the labia majora.


Type IV. All other harmful procedures to the female genitalia for non-medical purposes, for example pricking, piercing, incising, scraping and cauterization

To describe Type III mutilations more precisely, we have modified the WHO nomenclature, adding the extent of resection of the clitoris. “x” is used analogously to the TNM-classification system, implying that preoperative examination does not allow for a complete classification due to the overlaying scar after infibulation (see Figure 1).

### 1.2. Proposal for a Modified Classification of Type III Female Genital Mutilation

Type III:
oType IIIa. Partial or total removal and repositioning of the labia minora.oType IIIac. Partial or total removal and repositioning of the clitoral glans and of the labia minora.oType IIIb. Partial or total removal and repositioning of the labia majora.oType IIIbc. Partial or total removal and repositioning of the clitoral glans and of the labia majoraoType IIIc. Partial or total removal and repositioning of the labia minora and labia majoraoType IIIcc. Partial or total removal and repositioning of the clitoral glans and of the labia minora and labia majora.
Type IIIx: preoperative examination does not allow for a complete classification due to the overlaying scar after infibulation

## 2. Methods

### 2.1. Patients

The study was carried out at the Desert Flower Center (Waldfriede Hospital, Berlin). A total of 427 FGM patients were included in the evaluation. Patients were screened in a multidisciplinary team. After a comprehensive consultation and provision of thorough information on the anatomy and impact of FGM on urogenital and psychosexual health, treatment options were discussed. A total of 165 patients were found to be eligible for surgery.

The remaining 262 patients were either not considered appropriate for surgery, or opted for psychosexual counselling or attendance of our support group to exchange experiences with other FGM victims, considering surgery at a later stage.

During the first consultation of the patients presenting at our center, the patients’ current social situation, comorbidities, and general understanding of the anatomical and functional aspects of their mutilation were determined. Furthermore, the patients were thoroughly informed about the potential risks and individual benefits of the surgery in the context of a mutual decision-making process. It is important to make clear that surgery is not beneficial to all women who have undergone FGM [4,23]. Only in approximately half of our patients was surgery performed. Surgery was carried out if the mental status and social environment of the patients was stable enough to cushion the psychological burden of the surgical process, and if the potentially achievable surgical improvements were in accordance with the patients’ expectations. Therefore, the women with Type III mutilations presented in this study represent a large fraction of the patients who received surgery.

After being informed about the surgical techniques available, only 2 patients opted for only a deinfibulation procedure in case a resected glans was found underneath the scar during surgery. None asked for a partial deinfibulation.

All patients gave their written consent prior to surgery, if necessary in the presence of an interpreter. If the patients were minors, the legal guardian as well as the patient gave their written consent. In patients < 16 years (n = 2), only deinfibulation was performed due to menstrual/urination disorders.

The intervention was covered either by the patients’ health insurance or by our charitable foundation in the case that there was no coverage. No patient was rejected or had to contribute to the cost. Patients were usually referred by their gynecologist, other physicians, or patient advocacy groups or had sought treatment themselves.

Patients were categorized according to the WHO classification. Inclusion criteria for retrospective evaluation were all surgeries on FGM Type III. Small local flaps for improvement of the aesthetic outcome were included in the analysis; however, patients with additional larger reconstructive procedures were excluded from the evaluation to allow for comparability. All operative procedures were performed by the same senior surgeon.

The records of all patients with Type III mutilation were viewed retrospectively. The extent of clitoral involvement, operative time of prepuce reconstruction vs. no prepuce reconstruction, and postoperative complications of 85 patients with FGM Type III were retrospectively analyzed using patient records. Short-term complications were defined as complications within two weeks of surgery, while medium-term complications were defined as complications within six months of surgery.

### 2.2. Surgery

Surgery was performed under general anesthesia in order to prevent post-traumatic stress disorders, including flash-backs associated to the procedure. To allow for appropriate wound and psychological care within the first 1–2 postoperative days, patients remained in the hospital, which was reported as being appreciated by the patients. However, an ambulatory setting was also feasible, especially after simple deinfibulation procedures.

Briefly, the infibulating scar was released stepwise to visualize the true extent of the mutilation and spare the tissue for flap procedures. The further steps varied according to the findings as well as the wishes of the patient as defined preoperatively. In the case of a residual clitoral glans, either intact or buried in a cystic formation, the intervention was limited to a clitoral re-exposition similar to a deinfibulation procedure. When sufficient soft tissue excess was present, local flaps were used to construct a neo-prepuce [30].

In the case of a resected clitoral glans, the surgery was extended according to the suggestions of Thabet and Thabet and Foldès et al. (see Figure 2) [12,14]. In appropriate cases, the neo-prepuce was then constructed using the mobilized tissue as described above. After a nerve block using bupivacaine for post-operative pain relief, ointment dressings were applied. Patients routinely received NSAID until one week after discharge. No peri- or postoperative antibiotics were used. Patients were allowed to shower from the first postoperative day onwards.

### 2.3. Statistics

Student’s *t*-test for independent samples was used for statistical analysis of the operative time, while Fisher’s exact test was used for statistical analysis of postoperative complications, using GraphPad Prism 9.0 (GraphPad Software, San Diego, CA, USA). *p*-Values were rounded to 4 significant digits; *p*-values below 0.05 were considered statistically significant.

## 3. Results

Out of the 165 women who received surgery, 85 (51.5%) presented a Type III mutilation (Table 1), originating from 11 out of 24 countries and accounting for 20% of all women referred to us (Table 2). The mean age was 21.5 years (range: 14–59 years). The extent of the mutilation varied considerably, depending largely on the local customs and mutilation practices of each country. This was not only true for the infibulation procedure, but also for the extent of clitoral involvement. In 51 (60%) of the 85 patients with Type III mutilation receiving surgery, no clitoral mutilation was found when opening the infibulating scar.

The mean duration of surgery was 26.4 min. When re-elevating the clitoris, a significantly longer operative time (33.4 min) was found in patients with a mutilated clitoral glans compared to cases without clitoral mutilation (21.8 min, *p* < 0.0001; Figure 3; Table 3). In appropriate cases, a formation of the prepuce was performed using local flaps, extending the original technique of Foldès [14]. The operative time was 28.8 min on average in patients for whom a prepuce was reconstructed and 24.6 min in patients for whom no prepuce was reconstructed (*p* = 0.1693; Figure 4; Table 3). Thus, this procedure did not increase the time of surgery significantly, even though the extent of the dissection might have been somewhat larger in some patients.

No infections were seen. Two of the 34 patients (5.9%) with a partly resected clitoris required revision surgery, while none of the patients for whom an intact clitoris was discovered under the infibulation required revision. Differences in the rate of complications requiring surgery between patients with a partly resected clitoris and intact clitoris were not statistically significant (*p* = 0.1571).

Of the two women who required revision surgery, one woman complained of severe pain 10 days after a classical Foldès re-elevation of the clitoris, which was counted as a short-term complication. This technique, as well as the technique described by Chang et al. [24], includes sutures of the proximal scar to the periosteum in order to shape a sulcus covering the exposed clitoris. Releasing these sutures resolved the pain. The second patient presented 4 months after surgery with a hypertrophic scar surrounding the re-elevated clitoris (medium-term complication), leading to difficulties in hygiene and a substantial loss of clitoral projection. After scar resection, the clitoris was re-elevated and two z-plasties were performed to open the basis of the prepuce. No other prolonged stay or readmission was necessary.

Hypersensitivity and discomfort were reported at least until full epithelialization of the exposed clitoral stump, which took approximately 5–6 weeks. In two patients, hypersensitivity and discomfort took 4 and 6 months, respectively, to subside.

As reported in other studies, follow-up proved to be demanding [14,17]. Ten patients lived abroad and came to Germany only for surgery. Of the patients living in Germany, many were still in the process of their asylum application and thus lived in alternating facilities or were transferred to other cities. They often changed their cell-phone numbers and respective social workers, making follow-up even more difficult. We managed to follow-up with 53 patients (73%) with a mean of 24 weeks (range: 1 week–3.2 years). Of these, 40 patients (75%) presented in our outpatient clinic, while an additional 13 patients were contacted by phone.

## 4. Discussion

We found a significantly longer operative time in patients who presented with a completely or partially resected clitoral glans when compared to patients with an intact clitoral glans underneath the infibulating scar. Furthermore, we found a higher postoperative complication rate in patients with a mutilated clitoral glans. However, this difference was not statistically significant.

Along with an increase in numbers of women migrating from FGM-practicing countries and the growing demand for reconstructive surgery, expectations of surgical outcome have risen [4,17]. Until recently, procedures were limited to mere deinfibulation, which is undisputedly of great benefit to the affected women [13,31].

To date, there is only scarce evidence for the benefit of any kind of surgical or conservative therapy going beyond deinfibulation [4,10,11,17,18,19]. It is therefore not surprising that, due to the limitations of currently available evidence regarding the benefits of surgery and the potential risk of harming the residual clitoral function, the indication for reconstructive surgery beyond pain and fistula management is questioned [4,20,21,32].

Since deinfibulation and its obvious improvement in related functional impairments is widely accepted as beneficial, we selected the subgroup of Type III mutilations to identify additional surgical measures that could influence the outcome. In this context, we evaluated the WHO classification system with regard to its practicality to grade the extent of mutilation. Even though all women in this group were classified as Type III according to the current WHO classification, we found large differences within this group. For example, in Somali women, an intact clitoris was found underneath the infibulation in 65% of cases, regardless of the extent of labial resection (Table 1). It is important to notice that even though surgery extending beyond mere deinfibulation is not yet universally supported [4,20,21], there seems to be a strong demand for these procedures among the affected women. After being informed about the procedures at hand, only two patients opted not to proceed with clitoral elevation in the case of a clitoral involvement.

One of the limitations of our study was that our evaluation was limited to short- and medium-term complications, and did not assess the long-term complications. Furthermore, the effect of surgery on psychosexual health and other long-term health-related issues was not addressed. A better understanding of long-term results and individual benefits can only be obtained if validated evaluation methods and an extended classification system are established, enabling comparable studies. In this regard, an expanded classification is essential to correlate the extent of the pre-therapeutic findings with the results obtained. This could also serve as a useful tool in the selection of suitable patients for each specific intervention, including non-surgical therapies.

Another limitation was the fact that the functional outcome was not measured. It has yet to be analyzed as to whether a re-elevated clitoral stump has the same functional qualities as a re-exposed, unharmed clitoral glans. It can only be speculated that the long-term outcome will not be left unaffected by the extent of clitoral mutilation. Even though some studies describe an improvement in sexual function after releasing scar tissue and elevating the clitoral stump [14,24,27], it is likely that the degree of functional restoration will be more favorable in the presence of an intact clitoris.

In women with an intact clitoris underneath the infibulating scar, no major complications were found. The complication rate of patients with a mutilated clitoral glans was 5.9%, which was similar to the complication rate found by Foldès et al. (5.3%), who conducted a large study evaluating the outcome of clitoral reconstruction, mainly in Type II patients [14]. Furthermore, the operative time was significantly longer in patients with a mutilated clitoris. This is due to the need for an extensive scar tissue release, dissection of retaining ligaments, and stabilization of the clitoral body in its new position for re-elevation of the clitoris. The resulting larger surgical dissection was found to also influence the postoperative course. In cases where the clitoris and vulva were found to be almost unharmed underneath the infibulating scar, we noticed that the recovery time was much shorter and far less burdensome than in women who required a re-elevation procedure.

Making use of redundant tissue to reconstruct a neo-prepuce using local flaps whenever appropriate did not significantly change the operative time.

Throughout our study we were able to show that de-infibulation and re-exposition of the clitoral glans is a safe procedure, being associated with low morbidity, even when the clitoral glans is found in cystic formations that require additional small local flaps for a tension-free exposure [14,23,33]. The elevation of clitoral remnants after more extensive mutilations also proved to be safe, with no statistically significant difference in postoperative complications when compared to the group of patients with an intact clitoris. However, the presence of clitoral mutilation had a significant impact on the operative time with a not significant, but higher, need for revision surgery. In contrast, the reconstruction of a neo-prepuce did not have a significant impact on the operative time. In some cases, a worsening of the preoperative situation in terms of pain and sexual satisfaction has been reported [14,17]. Institutions involved in the operative care of FGM patients must be aware of this fact in order to provide appropriate structures and inform patients accordingly.

Since the degree of clitoral involvement is not easily assessable preoperatively, it is important to proceed stepwise under constant re-evaluation of the surgical site to avoid damage to potentially unharmed structures. Only specially trained surgeons who are familiar with all reconstructive techniques should perform this intervention in order to cope with unexpected findings. However, during pregnancy or labor, these considerations should not stop anyone from performing a deinfibulation. It is usually a simple procedure and might prevent multiple episiotomies or c-sections [31].

## 5. Conclusions

In our study, we showed that the presence of clitoral mutilation had a significant impact on operative time, with a higher, although not significant, need for revision surgery. We further showed that the reconstruction of a neo-prepuce did not have a significant impact on the operative time. Since the present WHO classification of Type III mutilations is based on preoperative findings only, we propose a modified classification system of Type III female genital mutilation, enhancing our understanding of the complex nature of Type III mutilations and serving as a useful tool to enable comparison of research studies and treatment outcomes.

## Figures and Tables

**Figure 1 ijerph-20-04439-f001:**
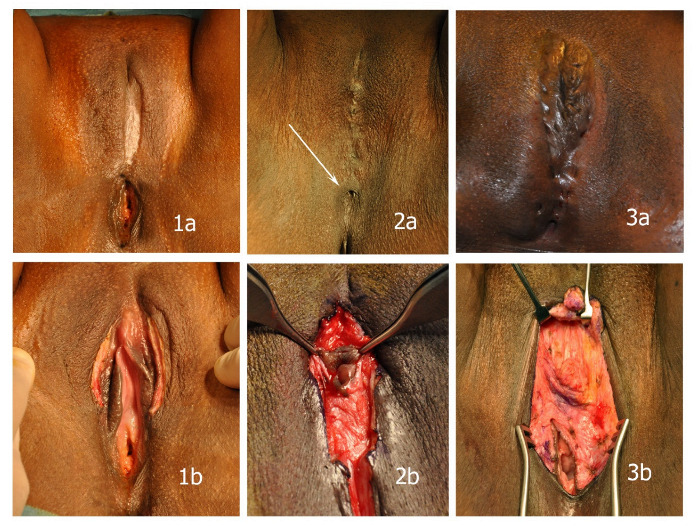
In spite of considerable differences in the extent of the mutilations, all cases shown are classified as Type IIIb mutilations according to the present WHO classification. (**1a**,**1b**) partial deinfibulation before sexual intercourse with almost unharmed minor labia and clitoris underneath the scar; (**2a**,**2b**) minimal opening left after infibulation (arrow) and unharmed clitoris in a small cyst encapsulated in scar tissue with resected minor labia; (**3a**,**3b**) extensive fistulae after suturing that required excision prior to reconstruction; resected clitoral glans and labia.

**Figure 2 ijerph-20-04439-f002:**
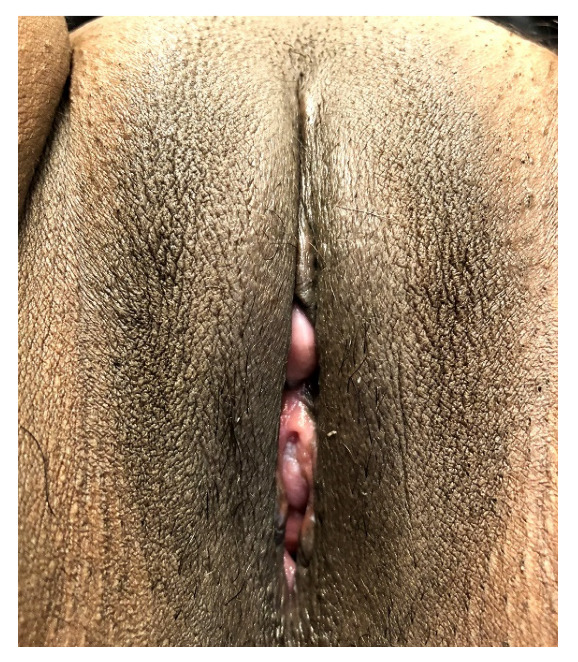
Three months after re-elevation of the clitoral stump and secondary epithelialization. Minimal vaginal showing without functional problems.

**Figure 3 ijerph-20-04439-f003:**
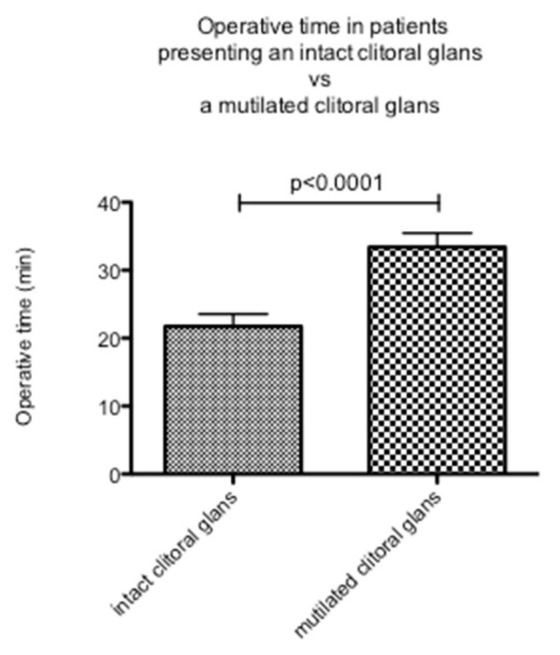
Operative time in patients with an intact clitoral glans compared to patients with a completely or partly resected clitoral glans.

**Figure 4 ijerph-20-04439-f004:**
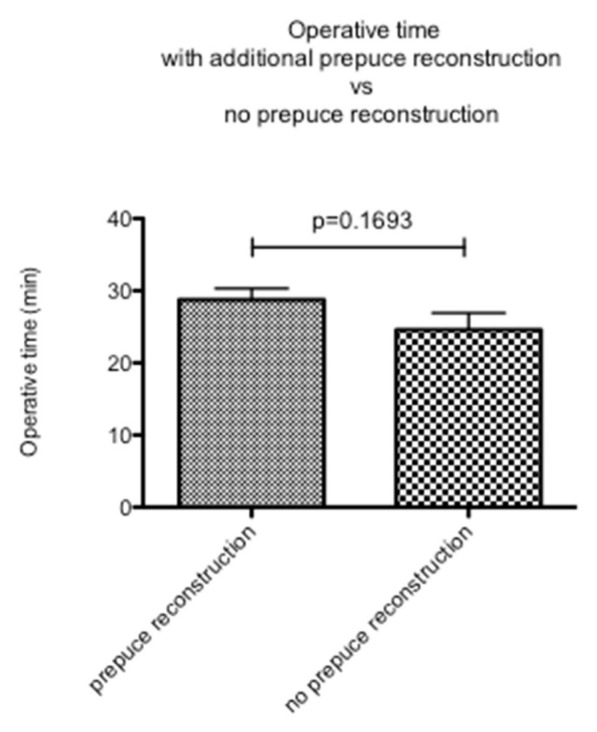
Operative time in patients with additional prepuce reconstruction compared to patients in whom no prepuce was reconstructed.

**Table 1 ijerph-20-04439-t001:** Number of patients with Type III mutilation according to our new proposal for a modified classification of Type III Female Genital Mutilation.

	n=	IIIa	IIIac	IIIb	IIIbc	IIIc	IIIcc
Somalia	65	26	13	6	2	10	8
Egypt	1	0	0	0	0	0	1
Ethiopia	3	1	0	1	0	0	1
Burkina Faso	1	0	0	0	0	0	1
Kenya	1	0	0	1	0	0	0
Eritrea	3	0	3	0	0	0	0
Djibouti	2	1	0	0	0	1	0
Sudan	4	0	1	1	1	1	0
Gambia	2	0	2	0	0	0	0
Sierra Leone	2	0	2	0	0	0	0
Nigeria	1	0	1	0	0	0	0
Total	85	28	22	9	3	12	11

**Table 2 ijerph-20-04439-t002:** Country of origin of all FGM patients referred and percentage of Type III mutilation.

Country of Origin	Number of Patients	Percentage of Type III Mutilation
Somalia	197	33%
Egypt	10	10%
Ethiopia	10	30%
Burkina Faso	9	11%
Kenya	15	7%
Eritrea	21	14%
Djibouti	5	40%
Sudan	12	33%
Gambia	11	18%
Sierra Leone	18	11%
Nigeria	16	6%
Other	103	
**Total**	**427**	

**Table 3 ijerph-20-04439-t003:** Mean duration of surgery in minutes (min) in patients with Type III mutilation according to the presence or absence of an intact clitoris and the additional formation of a neo-prepuce or the sole re-elevation of the clitoris without formation of a neo-prepuce.

	n=	Mean (min)
**Total of**	85	26.4
Clitoral resection	34	33.4
No clitoral resection	51	21.8
Formation prepuce	36	28.8
No formation prepuce	49	24.6
Clitoral resection and formation prepuce	19	32.4
Clitoral resection and no formation prepuce	15	34.8
No clitoral resection formation prepuce	17	24.7
No clitoral resection, no formation of prepuce	34	20.2

## Data Availability

Data is contained within the article. In case of specific interest, additional information is available on request from the corresponding author.

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
