# Peer review of "Postoperative Course of Reconstructive Procedures in FGM Type III-Proposal for a Modified Classification of Type III Female Genital Mutilation"

_ijerph, 2023, doi:10.3390/ijerph20054439_

Round 1

Reviewer 1 Report

In my opinion, this study is valuable because it adds new knowledge in the field of female circumcision with a scientific and acceptable approach. Reconstructive surgery helps to increase the quality of life of circumcised women. In fact, this study explains the unknown parts of this health problem. The methodology is well explained in this study and the results are convincing.

It is necessary to mention the ethical approach in the study. Also, the complications of the operation in the following months should also be taken into consideration.In general, I liked this article very much and I consider its publication valuable for the medical community and activists who work in this field.

Author Response

Dear Editors,

We would like to thank you and the referees for reviewing the manuscript “Postoperative course of reconstructive procedures in FGM Typ III - Proposal for a modified classification of Type III Female Genital Mutilation”.

Changes to the manuscript were marked up using the “Track Changes” function. We addressed all of the reviewer’s comments in a point-by-point manner and hope that the manuscript can now be considered for publication.

We look forward to your response!

Yours sincerely,

Reviewer 1:

Comments and Suggestions for Authors

In my opinion, this study is valuable because it adds new knowledge in the field of female circumcision with a scientific and acceptable approach. Reconstructive surgery helps to increase the quality of life of circumcised women. In fact, this study explains the unknown parts of this health problem. The methodology is well explained in this study and the results are convincing.

It is necessary to mention the ethical approach in the study. Also, the complications of the operation in the following months should also be taken into consideration. In general, I liked this article very much and I consider its publication valuable for the medical community and activists who work in this field.

We thank the reviewer for the valuable remarks. We agree that it is necessary to mention more information about the ethical approach in the study. Information about the ethics approval has been added to the revised manuscript:

Ethics approval:

The Ethics Committee of the University of Hildesheim (09.07.2018) and the Ethics Committee of the Board of Physicians Berlin (11.03.2021; Eth-59-20) approved the study. The design and performance of the study are in accordance with the Declaration of Helsinki.”

We described the complications of the operation in the following months in the following paragraph:

“Two of the 34 patients (5.9%) with a partly resected clitoris required revision surgery, while none of the patients in whom an intact clitoris was discovered under the infibulation required revision.

Of these two women, one woman complained of severe pain 10 days after a classical Foldès re-elevation of the clitoris. This technique, as well as the technique described by Chang et al. [1] includes sutures of the proximal scar to the periosteum in order to shape a sulcus covering the exposed clitoris. Releasing these sutures resolved the pain. The second patient presented 4 months after surgery with a hypertrophic scar surrounding the re-elevated clitoris, leading to difficulties in hygiene and a substantial loss of clitoral projection. After scar resection, the clitoris was re-elevated and two z-plasties were performed to open the basis of the prepuce. No other prolonged stay or readmission was necessary.”

Reviewer 2 Report

Dear Editor,

Kindly find my comments for the article titled:

Postoperative course of reconstructive procedures in FGM Typ III Proposal for a modified classification of Type III Female Genital  Mutilation

Attached are my comments

Kind regards

Reviewer

Reviewer 3 Report

Thank you for your submission

we read with interest the submitted manuscript; we have suggestions to improve the quality and the impact of the manuscript  

Overall English polishing is needed; there are parts of the paper that did not read well to me.

ABSTRACT  needs revision; the background seems to have incomplete sentences; they appear unfinished; the study type was missing, so as the location. The methods or parameters intended to be measured in the study were not defined. The conclusion was too long it was more like a discussion rather than a conclusion. Revision is needed.

INTRODUCTION.

Many paragraphs were without references [more than 10-15 lines] and were not supported with references. Revise, please.

You said that the aim [in the abstract ] is to define pre- and postoperative complications, while in the introduction, you seem to propose another classification of the FGM without defining any of the parameters of interest that you aimed to measure. kindly revise 

METHODS 

must be improved 

study type, duration; location of the hospital; its full name details were all missing. What about the ethical committee approval? Was that obtained prior to conducting the study?  what about patients' consent? we did not see anything in that regard throughout the paper. This is a critical point that should not be forgotten in this sensitive study content. 

[short- and medium-term outcome] is repeatedly repeated, yet we did not get the explanation of those terms; define the parameter of interest.

Figure 1; the photos need to be arranged in a more comprehensive way to be understandable

RESULTS 

the presented data were mostly descriptive; no correlation was done no P value was obtained; no comparison for the early and late complications; We would appreciate seeing more data.

DISCUSSION 

early in the discussion, we should highlight the study's main result? revise, please, .line 259-269. They belong to the introduction.

Lines 284-293 belong to the method section. 

There are many paragraphs without any references; support the discussion with more for example, 350-353

.Line 345 authors wrote [after extensive mutilation clitoris re-construction also proved to be safe] is that the author's opinion or?

study limitation needs to be revised and discussed in more depth

the implication of the current result on our practice in the field?

CONCLUSION 

must be revised into 4-5 concise lines.

Round 2

Reviewer 2 Report

Dear Authors,

I am now happy with how you have addressed the issues I raised in the previous version of your manuscript. You only need to tighten up the coclusion section and correct the grammatical errors in the entire manuscript.

Author Response

We thank the reviewer for his remarks. The remaining grammatical errors were corrected by a native speaker and the conclusion section was further tightened up:

“Throughout our study, we showed that the presence of clitoral mutilation had a significant impact on operative time, with a higher, however not significant need for revision surgery. We further showed that the reconstruction of a neo-prepuce did not have a significant impact on the operative time. Since the present WHO classification of Type III mutilations is based on preoperative findings only, we propose a modified classification system of Type III Female Genital Mutilation, enhancing our understanding of the complex nature of Type III mutilations and serving as a useful tool to enable comparison of research studies and treatment outcomes.”

Reviewer 3 Report

Dear authors 

we thank you for addressing most of the points raised during the first round 

the quality of the manuscript has indeed improved 

we thank the authors for the effort; well done 

we  have one comment 

in the discussion section, you explained your result and the implication of your proposed categorization system compared with another current one

have we not seen a comparison of the early and mid-term complications in your work compared to other studies? Especially since many surgical methods are described in the literature.  

Is there no earlier work that compared the complication of such surgery published earlier? Are you sure? 

Author Response

We agree with the reviewer that throughout the “Discussion” section, we need to compare our results with complication rates of earlier works. However, to our knowledge, no study has yet been conducted comparing the complication rate of reconstruction in patients with Type III FGM with a mutilated vs an intact clitoral glans underneath the infibulating scar.

Nevertheless, Foldès et al. conducted a large study evaluating the outcome of clitoral reconstruction, mainly in Type II patients (95% Type II, 5% Type III). Hereby, the authors showed that out of 2938 patients with clitoral reconstruction, 155 patients (5.3%) presented with postoperative complications [1], while throughout our study, 2 out of the 34 patients (5.9%) in whom a clitoral reconstruction was performed presented with a postoperative complication.

Thus, we added the following paragraph to the “Discussion” section of the revised manuscript:

“In women with an intact clitoris underneath the infibulating scar, no major complications were found. The complication rate of patients with a mutilated clitoral glans was 5.9%, which was similar to the complication rate found by Foldès et al. (5.3%), who conducted a large study evaluating the outcome of clitoral reconstruction, mainly in Type II patients. [1]

We thank the reviewer for this important comment.

Literature:

  1. Foldes P, Cuzin B, Andro A. Reconstructive surgery after female genital mutilation: a prospective cohort study. Lancet 2012; 380(9837):134-41